# Cultural Heritage for Sustainable Education Amidst Digitalisation

**Yianna Orphanidou** [iD], **Leonidas Efthymiou** *[iD] and **George Panayiotou**

School of Business, Department of Management, University of Nicosia, Nicosia 2417, Cyprus;
orphanidou.y@unic.ac.cy (Y.O.); panayiotou.g@unic.ac.cy (G.P.)
* Correspondence: efthymiou.l@unic.ac.cy

**Abstract:** The integration of cultural heritage in education facilitates critical thinking, experiential learning, cross-cultural collaborative learning and ultimately, quality learning experiences. This process is further enhanced by the increasing adoption of digital technology, which makes education more accessible. However, some countries in the European Union have low digital literacy and a high student dropout rate. Also, the use of cultural heritage in education is declining as young learners are becoming increasingly unaware of their cultural identity. Within this framework, a study of mixed methods (questionnaires and interviews) was conducted in three European countries to examine digital and cultural heritage competencies among young learners. The results of the paper reveal how digital cultural heritage increases learners' resilience by promoting competences for digital transformation, which in turn enhances learning and engagement with cultural heritage. Drawing on our findings, the paper proposes a new innovative hybrid model within the framework of sustainable education (SE).

**Keywords:** sustainability; cultural heritage; digitalisation; skills; education; hybrid teaching; sustainable education; sustainable learning

## 1. Introduction

### 1.1. Research Context

The current article explores how the embracement of cultural heritage in education through the adoption of digital technology can cultivate learners' skills and further contribute to sustainable education (Figure 1). Sustainable education (SE) refers to teaching and learning practices, skills and strategies which facilitate lifelong learning inside and outside the classroom [1]. According to Doukanari et al. [2], "research on sustainable education examines a wide range of learning practices, methods, and strategies, and how they consider, adapt to, and meet the diverse needs of student cohorts" (2021:1). The authors explain how SE has gradually expanded to comprise a wide range of practices and strategies, varying from sustainable feedback, students' sustainable development, problem-solving and hands-on experiences through to field trips, inter-disciplinary learning, internationalisation, sustainable curricula metrics, Multicultural Teamwork (MMT), Case-based Learning (CBL) and Problem-based Learning (PBL), among others.

According to Sterling [3], sustainable education (SE) can achieve an essential cultural shift. Cultural heritage learning fosters respect and understanding for cultural diversity, promotes intercultural discussion and contributes to more resilient and inclusive communities [4–6]. Cultural heritage refers to behaviours, beliefs, habits and artefacts that are passed down from generation to generation, forming a community's or society's identity. History, architecture, art, music, literature and language are all included, as are traditional knowledge, rituals and festivals [7]. Cultural heritage not only provides individuals and communities with a sense of pride and identity, but it also plays an important role in promoting intercultural discourse, protecting biodiversity and developing social cohesion.

Cultural heritage includes tangible cultural heritage and intangible cultural heritage. Tangible cultural heritage refers to physical artefacts created, maintained and passed down through generations in a civilisation. Intangible cultural heritage has been defined by UNESCO [8] as "the practices, representations, expressions, knowledge, and skills-as well as the instruments, objects, artefacts, and cultural spaces associated with them-that communities, groups, and, in some cases, individuals recognise as part of their Cultural Heritage". Oral traditions, performing arts, local knowledge and traditional skills are examples of intangible heritage.

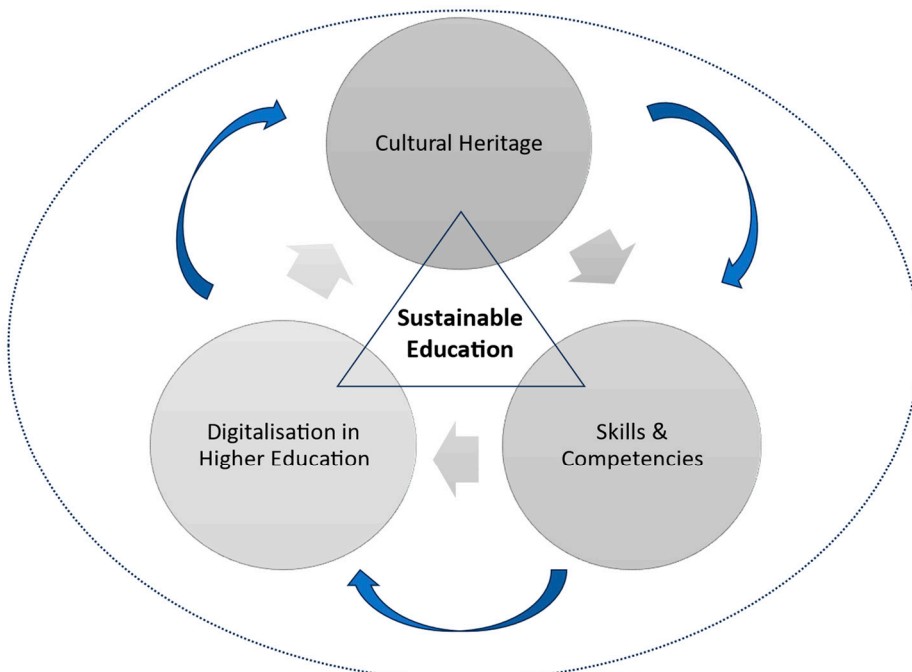

**Figure 1.** Literature review at the intersection of cultural heritage, skills and digitalisation as part of SE.

Within the framework of the Sustainable Development Goals (SDGs), set by the United Nations 2030 Agenda, it is critical to maintain and conserve cultural heritage for future generations to ensure its sustainability and relevance in an ever-changing world [9]. In light of increasing globalisation, cultural heritage began to decline. Young people became increasingly unaware of their cultural identity [10]. But lately, cultural heritage has gained popularity, along with public and scholarly interest around the world. Its conceptual reach can be seen in various Erasmus+ projects [11]. Also, cultural heritage is linked to urban sustainability [12]; preservation and revitalisation; experiences [13]; city regeneration [14]; and sustainable development [15], among others. Social scientists emphasise its functions in promoting ethnic, national and elite interests, while others highlight its creative and counter-hegemonic aspects. Promoting Education for Sustainable Development (ESD) in pursuit of the Sustainable Development Goals (SDGs) will strengthen collaboration with diverse sectors, particularly culture and science, to smoothly integrate ESD into the 2030 agenda. One main goal of ESD is to empower youth, prepare them to face the contemporary difficulties of unsustainable development and prepare them to be future decision makers. An aspect of the current study seeks to understand the level of youth awareness, attitudes and practices regarding tangible and intangible cultural heritage.

Moreover, as presented in Figure 1, the links between sustainable education, skills and cultural heritage are further enhanced by the increasing adoption of digital technology in education [16–19]. The inner set of arrows in the figure represent the interconnection and interdependence among the different components of sustainability. The outer set of arrows exists on the periphery of interdependence and reveals a dynamic in which the three

components are further enhanced and reinforced as part of a perpetual sustainability cycle. Individual skills and abilities can be strengthened through digitally aided education and training. The Institute for Prospective Technological Studies (IPTS), one of the European Commission's Joint Research Centres, has compiled a comprehensive study of national approaches to digital education policy around the world [20]. Recognising the importance of digital skills at the time, the European Parliament and Council of the European Union named digital competence as one of eight core competences required for lifelong (and sustainable) learning in 2006 [21]. Since then, the EU has developed numerous Digital Competence Frameworks (Dig Comp, DigCompEdu, DigCompOrg) to assist with the development of digital skills among all citizens, educators, educational organisations and consumers (DigCompConsumers). Four proficiency levels in five domains were developed, letting people evaluate their own digital skills and allowing comparisons between member states [22]:

a.   Information and data literacy;
b.   Communication and cooperation;
c.   Creation of digital material;
d.   Safety, and;
e.   Problem solving.

Human, digital and soft skills are more important in the twenty-first century than cognitive skills. They encompass abilities that robots and artificial intelligence lack or do not thrive on, but that people do have [23,24]. Learners with such talents will be in high demand since they can design and progress digital transformation [25], as well as contribute to societal advancement and innovation in general. Furthermore, the ability to manage change, notably resilience, adaptation, leadership and flexibility, is an important long-term ability for cultivating preparation for future advances [26]. In 2021, the European Union member nations had the lowest proportions of early school leavers. In contrast and contradiction with this, Italy (13%) and Cyprus (10%) reported the highest percentages. The EU member states have set themselves a target to reduce the rates of early school leavers to below 9% as the EU-level target by 2030. Sixteen member states have already met this EU-level target for 2030 for this indicator, including Lithuania [27].

The term "digital native" is increasingly being used in public discourse to describe generations of young people who have grown up surrounded by digital technologies. The term implies that young people intuitively understand how to use technology and thus do not require digital education or training. All EU digital policies during the last decade, including the Digital Agenda for Europe (2010) [28], the Digital Single Market for Europe (2015) [29] and a Europe fit for the digital age (2020) [30], have intended to make every European digitally competent. Although research on young people's usage of the Internet and technology in Council of Europe member countries is scant, Eurostat data provide some insight into the situation in the European Union. Consequently, 95% of young Europeans in 2021 aged 16–29 years reported using the Internet every day. However, the percentage of young people with a basic or advanced level of digital skills varies between 46% and 93%, with an EU average of 71%. Performing basic computer tasks, such as copying or moving a file or folder, is something, according to Eurostat [31], that 76% of all young people can do.

The use of digital technology has increased dramatically over the previous two decades. Digital technology is defined as "the use of electronic equipment to store, generate, or analyse data, as well as to promote communication and virtual interactions on social media platforms via the internet" [32]. Laptops, smartphones, computers, tablets and other similar devices are all considered electronic gadgets that are utilised for interpersonal connection, virtual communication and virtual engagement. Of course, research should consider not only the positive impact of technology but also its negative implications. Social media has swiftly changed the way young learners communicate with one another, igniting considerable scientific and public discussion over its possible impact on young learners socioemotional well-being and mental health. The necessity to bridge this knowledge gap has become more obvious in view of the COVID-19 pandemic [33]. For example,

Borthwick et al. [34] and Kumar et al. [35] state that "[l]earners can download the necessary information or upload their content using a plethora of digital resources". Web 2.0 tools (wikis, podcasts, blogs and so on) enable learners to create material, collaborate with others, evaluate each other's work and progress toward co-learning. The pandemic has forced people to rely on digital networks to preserve socio-emotional connections [36]. At the same time, most existing jobs will become obsolete due to technological advancements, and employees will require re-skilling and upskilling to expand their competencies and remain employed [37]. The use of technology and digital means in the education system has become increasingly important and necessary in order to meet the changing needs of students and provide them with a high-quality education that is accessible, flexible and sustainable [38].

### *1.2. Research Gap, Scope and Contribution*

The current article is part of the growing literature in the field of sustainable education (SE). The framework of SE does not solely contribute to sustainability and sustainable development. SE is a theoretical body on its own, which comprises a set of learning strategies, practices and pedagogies [2]. Adding further to the framework of SE, this is the first study to explore the interconnection of cultural heritage, skills and digitalisation and how they contribute to SE, as illustrated in Figure 1.

Also, the literature review revealed a need to explore additional learning methodologies for young learners. Even though young learners are progressively recognised as the fundamental stakeholders in the educational system, the vast majority of educational research continues to focus entirely on learners' viewpoints, positioning learners as passive information providers [39–41]. Young learners are more likely to be digital natives, meaning they grew up with technology and are more comfortable using it. Digital skills have implications for the future of the European workforce [42]. In an increasingly digital economy, those with strong digital skills will have a competitive advantage in the job market, whereas youth who lack such skills will find themselves in a position of disadvantage [22,25].

Drawing on the findings collected through mixed methods, this paper contributes to the literature with a new conceptual learning model, utilising tangible and intangible cultural heritage and emphasising the influence of digital cultural heritage as part of sustainable education. As outlined in the recommendations of the European Commission and the European Council [43], the introduction of this new innovative e-learning model that connects cultural heritage with digital skills is a new learning methodology that reflects the needs of digital native learners, with the aim of developing disciplinary and life skills and improving learners' key competences. This e-learning pathway can motivate learners and teens who are in danger of dropping out of school because it changes their understanding of and enthusiasm for digital technologies, such as social media and video games. In addition, the model considers the different needs, skills and competences of learners while adapting to their age, level of knowledge and abilities.

## 2. Research Design and Methodology

The research study was conducted in three European countries, Cyprus, Italy and Lithuania, between the years 2021 and 2023. The countries participating in the study were selected due to their striking similarity in terms of dropout rates, in line with the study's aim, which is to examine countries with low digital literacy and a high percentage of dropouts. Extensive secondary research was conducted to conceptualise the study by applying the method of a critical review [44]. The study included an in-depth examination of education curricula, national reports, European data from Eurostat, publications by the European Commission, the OECD and the Partnership for 21st Century Skills, and UNESCO studies. The purpose was to gather sufficient information on the three countries and their local educational systems and to demonstrate the extent to which education curricula have embraced cultural heritage elements. In terms of primary research, the study applied mixed research methods.

Questionnaires were used to gain a deeper understanding of young learners' digital and cultural heritage competencies. This quantitative approach was selected due to the need to measure the attitudes, opinions and characteristics of a large sample [45,46] and the need to collect a large amount of quantitative data from a sizeable sample [47,48]. The research population was composed of youth in private and public schools in Cyprus, Italy and Lithuania, as well as learners in tertiary education and youngsters that had dropped out of formal education and were more vulnerable in the labour market. An online structured self-administered questionnaire consisting of fifteen questions was used to gather data, covering areas such as demographics, familiarity with digital means and level of competence in relation to cultural heritage. The study adopted probability sampling. The sample was drawn from each institution's list of learners (sampling frame). The collection of quantitative data was conducted online through Google Forms. The questionnaire link was shared by each institution participating in the study. In total, 820 questionnaires were collected. The responses were analysed through SPSS (Statistical Package for the Social Sciences, Version 21). The questionnaire is available in the Supplementary Material (Questionnaire S1).

To avoid biased responses and to alleviate respondents' concerns or reluctance to participate in the current survey, the respondents were assured in advance that information generated from completed questionnaires would be anonymous and completely confidential and would be used only for the academic purposes of the current investigation. A cover statement on Google Forms aimed to explain to the respondents the research topic, aim and objectives, so that they could understand the crucial importance of their contribution prior to agreeing to respond to the questionnaire.

Also, interviews were conducted to ascertain the views of key stakeholders. The interviews' participants were key stakeholder representatives, including museum officers, policymakers and education authorities. The fieldwork's aim was to grasp the opinions of different authorities who have a role to play at the intersection of cultural heritage, digitalisation and education, and more specifically, with regard to the skills and competencies that future graduates should be equipped with. The interview method was used to facilitate the exchange of information between the researcher and the respondents since the research question required a detailed analysis on the part of the interviewees and thus demanded a method capable of providing in-depth and exhaustive information. Interviews were therefore deemed the most suitable method since they provided interviewees with plenty of freedom to articulate their thoughts and present their opinions. In the qualitative part of the research, purposeful sampling was applied since this technique is commonly used in qualitative research and allows for the optimum use of limited resources [49]. This entails locating and selecting individuals or groups of individuals who are particularly knowledgeable about or experienced with the phenomenon of interest [50]. Twenty-one (21) semi-structured interviews were conducted with educators, policymakers and representatives of cultural identities from the three selected countries. The interviews were conducted in native languages and translated into English (which is the project's official language) by the project's designated translator.

The interviews were analysed manually through two-cycle coding [51], as presented in Table 1 below. The criteria used for coding are the 13 competencies that appear in Figure 2 later in the analysis. As mentioned earlier, the purpose was to explore the preferences of different authorities with regard to the skills and competencies that future graduates should be equipped with. The first cycle of coding included a review of field notes. This process was undertaken immediately after each interview using a "data-set sheet". Reviewing the findings right away was helpful in recalling information that may have slipped the note-taking during a fast discussion. The first cycle included categorising and labelling officias' responses. It was also about formulating an interpretation since different authorities had different expectations about the skills and competencies expected from future graduates and employees. In other words, the process was about interpreting expressions and synthesising multiple sentences, which then became small sentences.

**Table 1.** Phases of two-cycle coding and meta-coding.

| | Phases |
|---|---|
| **1st-Cycle Coding** | - Undertaken immediately after each interview using a "data-set sheet" <br> - Review of field notes <br> - Categorising and labelling official responses <br> - Formulating interpretations <br> - Developing small sentences |
| **2nd-Cycle Coding** | - Further analysis and re-organisation of material <br> - Synthesisation of sentences into paragraphs |
| **Meta-Coding** | - Development of longer, analytical pieces of text <br> - Integration of paragraphs into the article's analysable units. <br> - Linking of analysis back to theory |

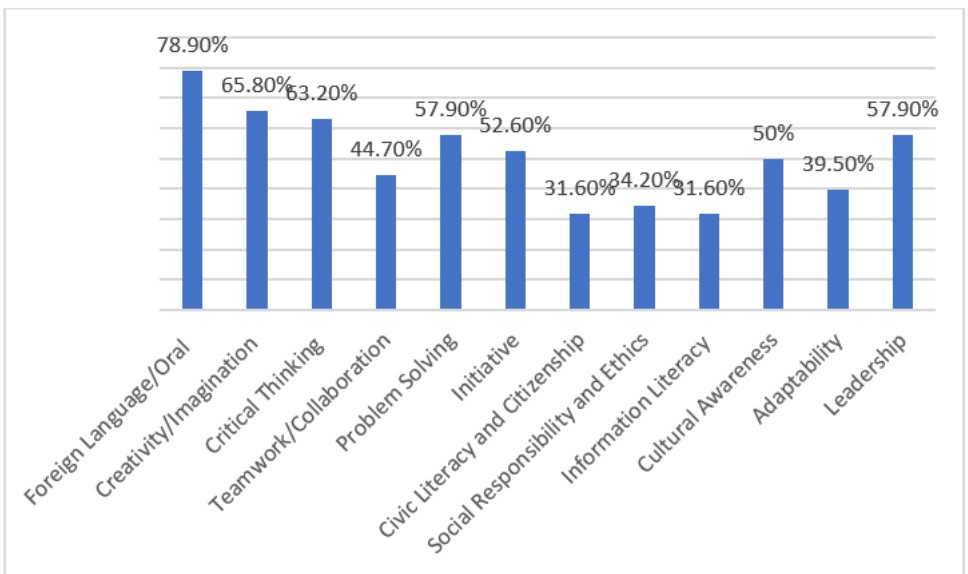

**Figure 2.** Competences' importance in enhancing employability.

Then, the second-cycle coding enabled the researchers to re-analyse, re-organise and resynthesise the material produced through the first-cycle coding to produce bigger chunks of analysis. Then, the "meta-code" method [52] was employed to develop longer, analytical pieces of text to be integrated into the article's analysable units. Towards the end of the study, when a substantial part of the article was written, the second-cycle phase became redundant. It was replaced by "meta-coding", which included direct editing of the article's analysis. Throughout the research collection process, the findings were linked back to theory and are presented in the "Findings" section below.

Since the study was undertaken as part of an EU-funded project, research ethics approval was granted by the project consortium's legal office. The collection of findings through interviews and questionnaires applied international research ethics principles and tools, including informed consent, anonymity, confidentiality and data privacy. The participants (for interviews) and respondents (for questionnaires) were provided with a cover letter explaining the aims and objectives of the study, along with potential benefits for education.

Finally, the use of mixed methods allowed the researchers to fully explore the status of the competence of youth in digital cultural heritage and the readiness of current education and cultural providers to deliver digital cultural heritage education, focusing on enhancing the skills and competences of young Europeans to enhance their employability and entrepreneurial capability.

## 3. Data Analysis and Results

The questionnaire sample consisted of 239 questionnaires from Cyprus, 458 questionnaires from Italy and 123 questionnaires from Lithuania. The gender representation of the sample was 50.3% female and 48.1% male. The sample that responded was equally distributed between the genders, with a slight predominance of the female gender. Most of the respondents resided in their country of origin, with insignificant percentages attributed to other origins. The largest percentage of respondents belonged to the 14–16 age group (58.8% of the sample), followed by the 17–19 age group (41.1%), and lower percentages are to be found for the 20–25 and 26–30 age groups. Furthermore, 87% of the sample had primary-to-secondary education, with 2.4% representing dropouts and 7.3% having university education. Finally, only 3% of people had pursued but never completed university education.

### 3.1. Competencies and Digital Means

The first part of the questions aimed to identify the competencies that the young respondents valued as most important. The survey's respondents had to select from a list of thirteen competencies that had been identified as the most important ones by the OECD, the European Commission and the Partnership for 21st Century Skills [53]. As presented in Figure 2, the data analysis revealed that foreign languages (78.9), creativity (65.8) and critical thinking (63.2) were valued as the top three important competences for enhancing employability, with problem solving and leadership following at 57.9%. All thirteen competencies had a significant percentage of 30% or above, which indicates equal importance. The replies of the young respondents reveal a high level of awareness of the competencies they need to possess to enhance their employability (see Figure 2).

As the research focuses on digital natives' skills, it was important to identify what type of digital devices young people use most often. Smartphones are by far the most commonly used digital device among young people, with 98.9% of the sample selecting them as their first choice. Second in line are laptops and PCs at 44.9%, followed by tablets at 28.3%. The Mascheroni and Cuman [54] study supports that in European countries, young people go online using multiple devices. It has been determined that young people prefer to use the web for social networking, gaming, and chatting [55]. Overall, young people today use a wide range of digital devices for a variety of purposes, and the types of devices they use can vary depending on many socioeconomic and cultural characteristics.

Due to the need for a larger screen and more powerful processing, young people frequently use laptops and tablets for studying, gaming and other activities. Wearable tech, smartwatches and other wearable devices are becoming more popular among young people for communication, fitness tracking and other uses. While older teens (aged 17–19) preferred laptops and desktops, the younger respondents (aged 14–16) appeared to be more likely to use tablets or smartphones.

The advancement in technology, and especially the introduction of social media such as Facebook and Instagram, which affect the way we live, work and, more importantly, learn, have changed people's lives dramatically. Teachers and professors are increasingly incorporating social media into their classes, whether they are online or in person, to engage students and advance their knowledge. Changing pedagogical approaches and implementing new teaching strategies, organising and controlling learning, and accessing important information sources have all benefited from a technology-enhanced learning environment [56–59]. In summary, social media is affecting and moulding how young learners' study and interact today, and many educational institutions and organisations have developed online courses and e-learning platforms that provide educational content in a variety of formats, such as video lectures, online quizzes, and interactive activities. The data analysis showed that among users between the ages of 14 and 19 who utilise digital methods to access learning and general information, 73.5% of the overall sample ranked YouTube as their top option (see Figure 3), with this being consistent with the most comfortable platform used for learning (see Figure 4). It is extremely intriguing that

e-books and PowerPoint presentations, which are widely used in formal education, are not preferred digital media for people between the ages of 14 and 30 (see Figures 3 and 4). A significant result of this research was the requirement to redesign pedagogical frameworks for online learning in education.

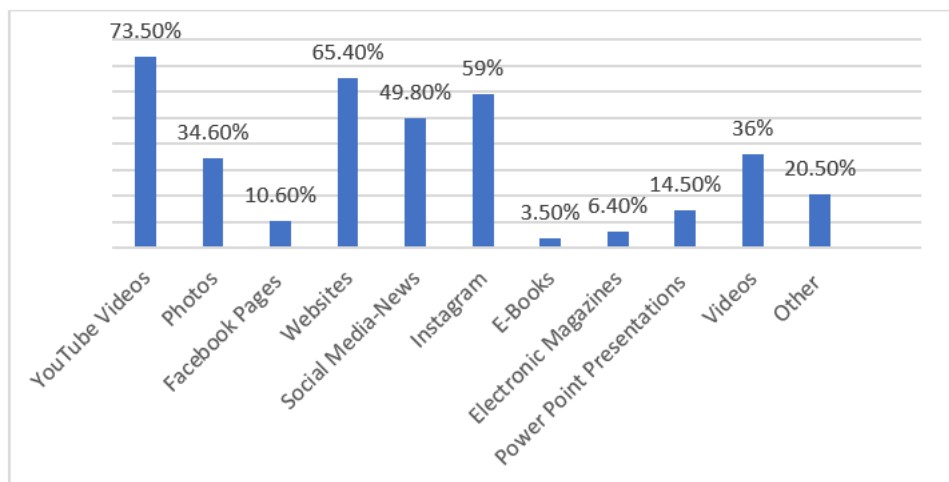

**Figure 3.** Use of digital means for learning and information purposes.

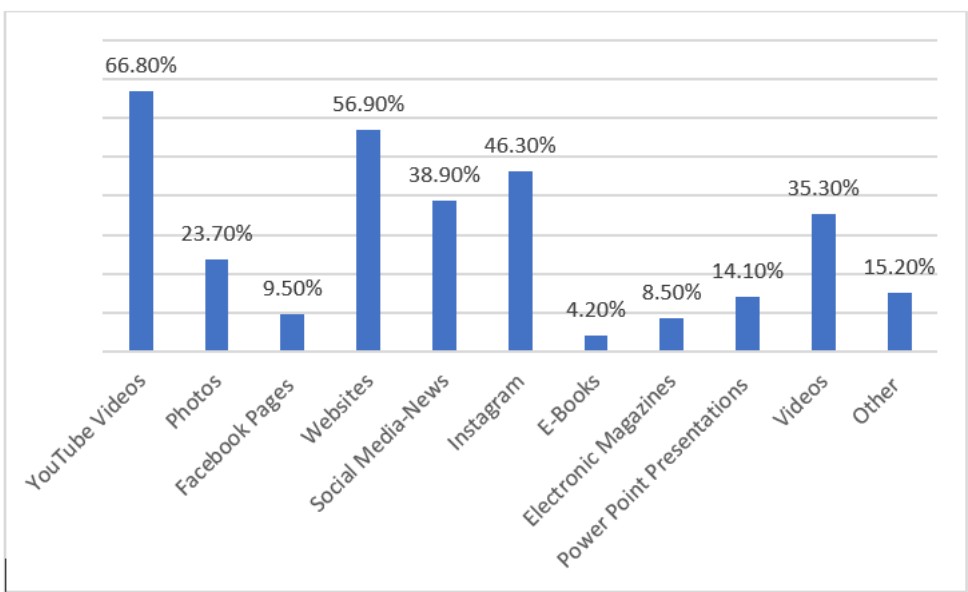

**Figure 4.** Digital tools learners feel more comfortable using in learning.

### 3.2. Cultural Heritage

As described in the Introduction, UNESCO [7] classifies cultural heritage into three types: tangible, intangible and natural. An evaluation of the literature suggests that cultural heritage is an important component of our cognitive knowledge [60] and should be taught in schools. In the 1990s, Bruner [61] and Wertsch [62] wrote stirring papers on the tradition of cultural psychology, stressing the fact that culture is entirely fabricated and that it shapes and allows the functioning of the human mind. Their view was that learning and thinking always occur in specific cultural contexts. "Culture shapes the mind of an individual. Its individual expression is achieved through the creation of meaning, through the attribution of meaning to things in different contexts and situations" [63]. Among the aims of this study was to determine how knowledgeable the young respondents were about tangible and intangible heritage.

As presented in Figure 5, there is a lack of awareness around both tangible and intangible cultural heritage since the respondents struggled to identify all nine of the assessed cases as cultural heritage. From the whole sample of respondents in the study, only 30% identified the nine assessed cases as most relevant to tangible and intangible cultural heritage. The study's findings confirm the importance of emphasising cultural heritage in curricula because failing to do so puts pride and respect for European identity in jeopardy.

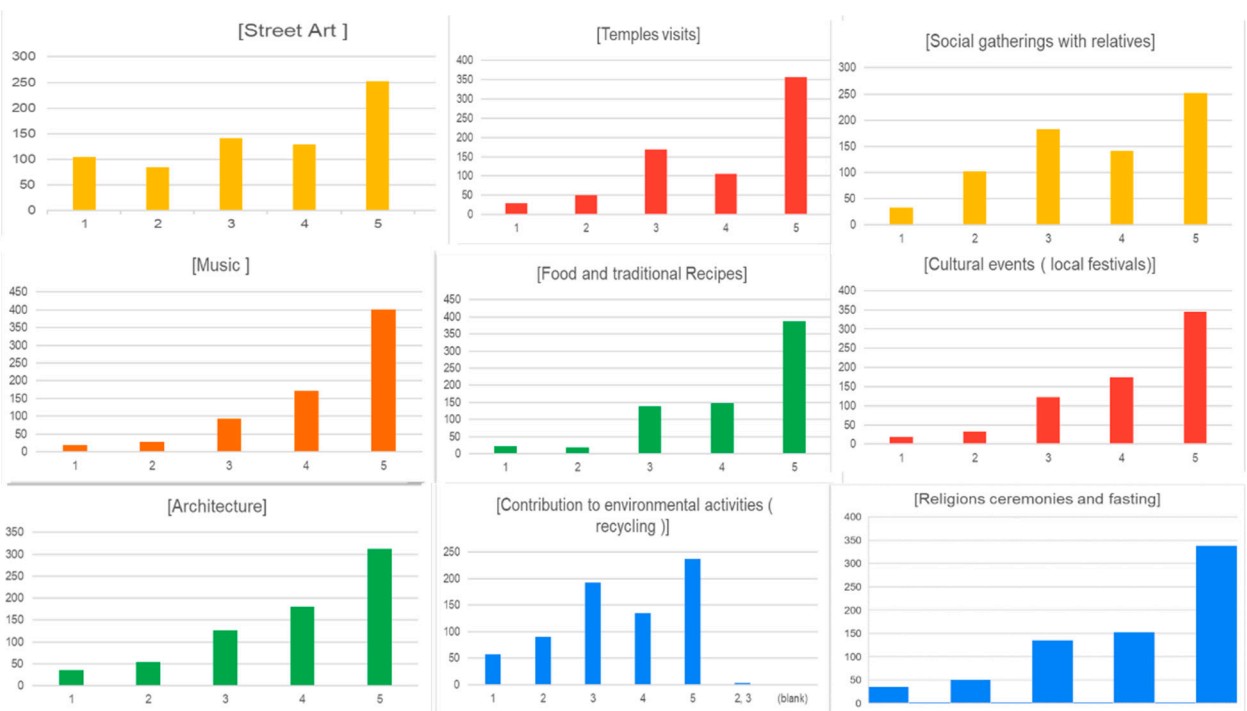

**Figure 5.** Identify the following examples in relation to cultural heritage, with 1 being least relevant and 5 being most relevant.

### 3.3. Education and Digital Cultural Heritage

The interviews provided a significant data pool, as the interviewees were purposefully selected to represent education and cultural providers in the three countries. From the analysis of the data collected, there is a consensus in so far as the way cultural heritage is taught in primary education, which involves mostly courses such as music, art, geography and religion. In secondary education, there is an emphasis on languages, history, economics and civic education. There is, however, a significant gap in how education systems define cultural heritage. From one country and language to another, the terms "culture", "cultural heritage" and "education" were not defined in the same way. There was agreement that "cultural legacy", which includes both tangible and intangible elements, has an impact on both the past and the present.

From the interview scripts' content analysis, there was a strong agreement that heritage and education should be seen as tools for sustainable development rather than just a reaction to the market-based economy. "Cultural heritage is not a "duty" or an encounter with heritage, but a tool that in the right hands can give good results" [11]. Within this context, it is imperative that education, including its primary objectives and strategies, be re-considered, including issues pertaining to digital cultural heritage education. The use of digital cultural heritage education may enhance the development of soft skills and competencies necessary to create resilience in European youth.

We can also increase learners' resilience in the cultural sector by holding various thematic workshops in open spaces (e.g., museums, archaeological parks, nature parks, national parks). "Workshops will affect the acquisition of knowledge and skills, or their

consolidation, and thus learners will be more resilient". Cultural heritage education thus enhances people's ability to become not only fulfilled citizens able to live in society but also responsible citizens regarding the protection of cultural heritage. The use of digital cultural heritage education may help to improve the soft skills and competencies required to generate resilience in European youth. "Learning about belonging to our society and community access is a must as well as for our cultural identity in order to promote social engagement and active participation in society".

From the interviewees' analysis, the authors gathered very strong statements that support the need for the utilisation of cultural heritage in education curricula. The respondents' repetitive feedback on the benefits of using cultural heritage in education provides a strong basis regarding the need for a new pedagogical model.

## 4. Discussion

### 4.1. Innovative Hybrid Educational Model

SE cannot become fully sustainable without integrating aspects of cultural heritage into the learning process. The current paper suggests a "Digicult" model (Figure 6), which emphasises the use of cultural assets in the learning experience to improve learners' skills and competencies. The name "Digicult" comprises the word's digitalisation and culture. Based on this model, learners develop information, intellectual abilities and a broader variety of competences on themes such as cultural heritage maintenance and societal well-being by actively experiencing or analysing elements of cultural heritage. This type of knowledge leads to long-term economic growth initiatives, including chances for respectable work. The suggested model ensures inclusiveness for young learners aged 14–30 while taking into consideration various educational backgrounds and motivating them to engage in lifelong learning. The model is appropriate for formal, non-formal and informal education.

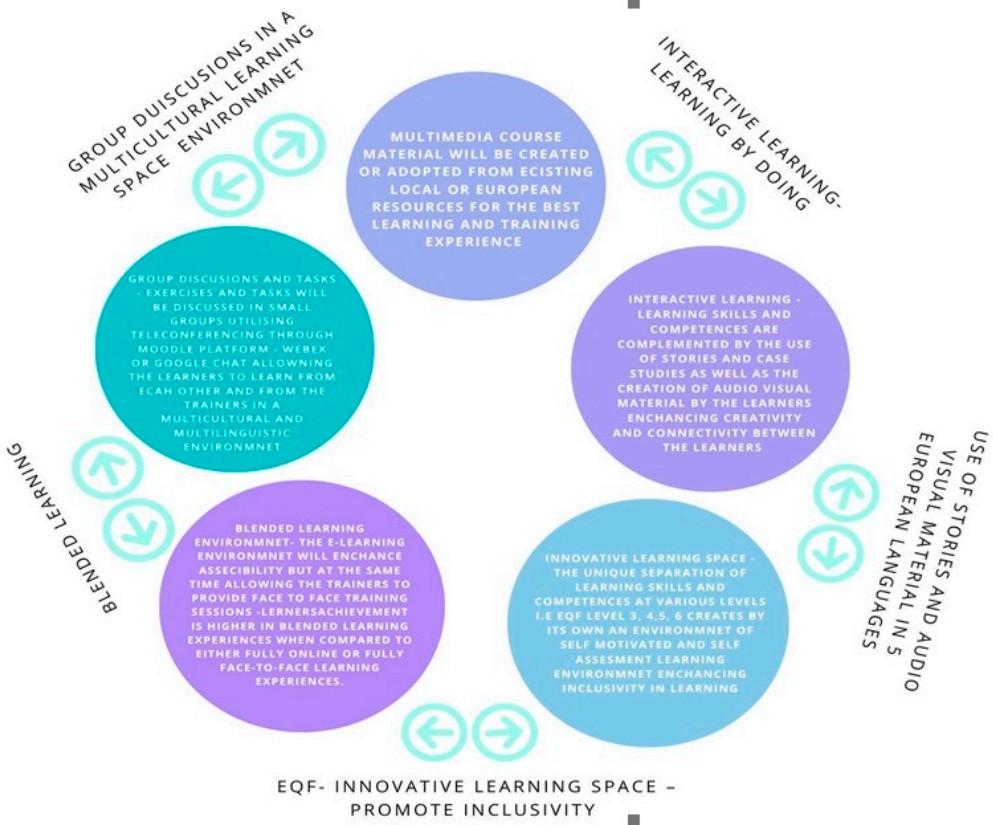

**Figure 6.** Digicult model for hybrid education.

Moreover, the competencies revealed by the analysis of questionnaires and interviews emphasise the need for enhancing foreign languages, written and oral communication, creativity/imagination, critical thinking, problem solving and the ability to work as part of a team. The diversity of the audience revealed the need for an innovative curriculum that takes into consideration individual learning needs as well as creating an environment of inclusivity. The model focuses on enhancing skills and competencies in a digital era by utilising cultural heritage. Learners develop information, intellectual abilities and a broader variety of competences on themes such as cultural heritage preservation and societal well-being by personally experiencing or analysing cultural assets. This kind of knowledge leads to sustainable economic growth actions, including opportunities for employment. Also, the research revealed that an e-learning pathway does not necessarily need to be 100% digitally delivered, as young learners' digital skills vary from country to country. The option of designing a blended mode of delivery may be more appropriate since it is likely to lead to better learning results.

As presented in the model, several learning strategies are utilised, such as story-telling, multicultural work, reflective tasks, teamwork and continuous assessment. Also, all these strategies can be utilised both face-to-face and online. In addition, they can be applied to individual tasks (self-assessments) and/or tasks involving teamwork (collaborative tasks). Nevertheless, it is important to underline that engagement and interaction among students are fundamental. Also, these strategies can be applied for both summative and formative assessments.

Moreover, "learning by doing" is applied as a means of facilitating the active involvement of learners in the learning process [16]. In other words, the model provides the opportunity to learn through concrete experiences and the application of what has been learned in a real-life situation—either individually or as part of a team. The ongoing process of the assimilation of experience into knowledge, known as Kolb's learning cycle, involves an interaction between action, reflection, experience and abstraction [64]. The four stages of Kolb's learning cycle are concrete experience, namely reflective observation, abstract conceptualisation, active experimentation and the foundations of experiential learning. In general, concrete experience is a time when learners engage in an experience in order to learn. Learners review their experiences through reflective observation. Departing from Kolb's learning cycle to the Experiential Learning Theory, the paper proposes a hybrid innovative model while adopting a pedagogical approach to implementing experiential learning in a digital learning environment for the education and training of young learners as part of SE.

Kolb's learning cycle can be utilised for reflexivity while attempting to apply the Digicult Model as a novel framework for learning. Many of the strategies presented in the Digicult Model (Figure 1) exist within the framework of reflexive learning. Through scholar–learner and learner–learner in-class collaboration (virtual or physical), reflexive learning provides space for the re-invention [65] of cultural heritage and the development of cultural identity. Reflecting on experiences has a central role in learning. The suggested model provides a context that fuels and is fuelled by the curiosity to search for, revitalise and merge traditional components of cultural heritage with contemporary, socially constructed learning. By reviewing and reflecting on cultural experiences, the Digicult model suggests a change in abstract knowledge to practice, cultivating a more systematic integration of cultural heritage in education.

However, reflexive learning can also be used at higher levels of decision making as a mode of inquiry and repositioning. Stemming from the interviews, our findings suggest an inconsistency in how cultural heritage is defined among different education systems. From one country and language to another, the terms "culture", "cultural heritage" and "education" are not defined in the same way. Thus, at a higher level, reflecting on current experiences and practices can facilitate a dialogue between stakeholders about what cultural heritage is and how it is utilised in a local context. According to Gorli et al, [66] reflexive learning can be used as a basis for action, questioning the status quo and seeking

change. Likewise, through cross-boundary collaboration, reflexivity can facilitate a better understanding of how other EU countries and regions understand cultural heritage and how it is currently utilised in education. Eventually, reflexive practice may result in a re-configuration of "cultural heritage" and its use in education. In this sense, reflexivity does not only facilitate performance and creativity, but it also acts as a transformative power that is likely to enable new possibilities, new understandings and clarity on courses of action through co-creation and inter-organisational collaboration [65].

As discussed in the Introduction, the term "digital native" implies that young people intuitively understand how to utilise technology, and therefore, they do not require digital education or training. All EU digital plans during the last decade, including the Digital Agenda for Europe [28], the Digital Single Market for Europe [29], and a Europe fit for the digital age [30], have attempted to make every European a digital native. However, research on young people's usage of the Internet and technology in Council of Europe member countries is limited. Eurostat data provide some insight into the situation in the European Union. In 2021, 95% of Europeans aged 16 to 29 reported using the Internet every day. The proportion of young people with basic or above-basic digital skills spans from 46% to 93%, compared to the EU average of 71%. In addition, 76% of all young people said they had carried out basic computer tasks like copying or moving a file or folder. At the same time, previous studies suggest that some young people are not as savvy (or unsavvy) with digital technology as we might think [67]. While they might not be technophobes, they still may not have certain literacy skills when it comes to digital devices, or they may be digitally deprived [68]. According to Eurostat [27], digital resources can offer valuable learning opportunities and life-changing experiences for students in a range of academic fields, especially those in subjects like hospitality and tourism.

### 4.2. Implication to Practice

While the practices discussed earlier are part of SE, what is of great importance in this model concerns learning opportunities based on substantial historical and/or cultural backgrounds, allowing students to become more deeply involved in their studies or even to recognise themselves for the first time as unique cultural scholars. This is because the model utilises digital cultural heritage while focusing on skills and competencies such as critical thinking, creativity and innovation through the learning of cultural heritage (tangible and intangible). This is the reason we argued that SE can never be fully sustainable unless it integrates cultural heritage experiences into the learning process. Also, according to the e-learning education paradigm, new digital tools and content are required to engage young learners to develop critical core competencies that will increase their employability and productivity. Learning does not have to be online. It can be blended learning, given that teaching in brick-and-mortar environments can still incorporate computer-based tasks and interaction. The implications for education are considerable since the use of a model that places more emphasis on interactive outputs than on content can support the design of interactive labs (physical or online) that cover both the acquisition of new digital skills and the development of knowledge and abilities that will unite young people in Europe through a digital cultural environment. This is the essence of SE, which brings together learning strategies and pedagogies for resilience, inclusiveness and progress.

**Supplementary Materials:** The following supporting information can be downloaded at: https://www.mdpi.com/article/10.3390/su16041540/s1, Questionnaire S1.

**Author Contributions:** Conceptualization, Y.O. methodology, G.P. collection of findings, L.E., analysis of findings, L.E. and Y.O.; G.P., concluding discussion, L.E., review and editing. All authors have read and agreed to the published version of the manuscript.

**Funding:** This research received no external funding.

**Institutional Review Board Statement:** The study was conducted in accordance with the Declaration of Helsinki, and since the study was undertaken as part of an EU-funded project, research ethics approval was granted by the project consortium's legal office.

**Informed Consent Statement:** Informed consent was obtained from all subjects involved in the study.

**Data Availability Statement:** Data will be destroyed after publication in line with International Research Ethics' standards and as outlined in the Informed Consent Form.

**Acknowledgments:** This paper was supported by research conducted under the project titled DIGICULT under the 2021-1-IT01-KA220-VOCATIONAL-000034836 Erasmus+ KA220-Cooperation partnerships in education and training.

**Conflicts of Interest:** The authors declare no conflicts of interest.

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
