# Peer review of "Cultural Heritage for Sustainable Education Amidst Digitalisation"

_sustainability, doi:10.3390/su16041540_

Round 1
Reviewer 1 Report
Comments and Suggestions for Authors
This is an interesting and broad piece of research and I commend you for a thorough literature review! Here are my suggestions for improving the manuscript:
1. Provide a reference(s) to support your statement on lines 48-49,
2. Provide concrete examples to support your statement on lines 110-111,
3. Remove final paragraph ahead of Section 2(lines 153-157),
4. Expand on methodology section starting with lines 178-182 - how did you collect the quantitative data - face-to-face or online? How was your sample selected? Did you do follow-up requests to provide data?
5. You used qualitative content analysis - did you follow the recommendations from a known author?
6. I suggest comparing your results from Table 1 with that of industry leaders across the EU. What do the believe are the top 3-5 competences?, and
7. For your statement on lines 252-254, provide evidence that teachers are modifying their classroom instruction to include more technology.
Comments on the Quality of English LanguageI only found minor punctuation errors on lines 88, 125,171 & 387. On line 162, you use STUDY twice in the same sentence, replace one with a different word. On line 168 replace qualitative with quantitative. I suggest you reword lines 170 - 172 so it is one, complete sentence. ".....characteristics of a large sample [30, 31], and the need to collect a large amount of quantitative data [32,33]."
Author Response
Dear Reviewer,
Thank you for your suggestions, which aim at strengthening our article. We found every single suggestion useful and therefore, we made an effort to implement all suggested changes (please see a list below). In the journal’s guidance, there was a note, asking us to highlight our changes and revisions in the analysis. However, with the additions of news parts; extensive editing throughout the script; and proofreading at the end, almost every single line had to be highlighted. Therefore, instead of highlighting changes, we decided to submit a table with thorough and detailed explanations of what we did.
Once again, thank you very much for accepting to evaluate our revised work. If any other changes are needed, we are more than happy to work on the paper.
We are much obliged for your kind input.
Cordially,
Reviewer 1 |
Actions |
1. Provide a reference(s) to support your statement on lines 48-49. |
Many thanks for this suggestions. The entire paragraph has been revised; and examples and sources have been added (50, 56, 57, 58, 59). |
2. Provide concrete examples to support your statement on lines 110-111, |
Thank you. Examples have been provided, along with their references. |
3. Remove final paragraph ahead of Section 2(lines 153-157), |
The paragraph has been removed. Thank you. |
4. Expand on methodology section starting with lines 178-182 - how did you collect the quantitative data - face-to-face or online? How was your sample selected? Did you do follow-up requests to provide data? |
Many thanks for this comment. The following analysis has been added:
The study adopted a probability sampling method. The sample was drawn from each institutions’ list of learners (sampling frame). The collection of quantitative data was conducted online, through Google Forms. The questionnaire link was shared by each institution participating in the study.
In addition: following yours and other reviewers’ feedback, we added many other details explaining the research design, including sampling approach, sampling frame, saturation et cetera. The ‘Methodology’ section has been restructured entirely. Thank you for your suggestions, which at strengthening our article. |
5. You used qualitative content analysis - did you follow the recommendations from a known author? |
No qualitative content analysis was applied for the literature review. What we meant with that phrase is that we analysed our qualitative findings. Therefore, this phrase has been removed, to avoid confusion. What we meant is that we applied coding and categorization of qualitative findings. To do so, we applied a two-cycle coding. A relevant section has been added in the analysis. |
6. I suggest comparing your results from Table 1 with that of industry leaders across the EU. What do the believe are the top 3-5 competences?, and |
The participants in the survey had to select from a list of thirteen competencies that had been identified as the most important ones from OECD, the European Commission and the Partnership for 21st Century Skills [36] |
7. For your statement on lines 252-254, provide evidence that teachers are modifying their classroom instruction to include more technology. |
Thank you. Four new resources have been added to support this statement [46-49] |
Comments on the Quality of English Language I only found minor punctuation errors on lines 88, 125,171 & 387. On line 162, you use STUDY twice in the same sentence, replace one with a different word. On line 168 replace qualitative with quantitative. I suggest you reword lines 170 - 172 so it is one, complete sentence. ".....characteristics of a large sample [30, 31], and the need to collect a large amount of quantitative data [32,33]." |
Thank you ever so much for suggesting direct changes in the script. We applied all suggestions; and also sent the script for professional proofreading. |
Reviewer 2 Report
Comments and Suggestions for Authors
Dear Authors,
key tables must be improved to be visible and presented clearly. Until this, I
can't recommend it for publishing.
Moreover, you write a scientific work, not a best-seller. Try to avoid
sentences that do not say much to understand your work.

Author Response
Dear Reviewer,
Thank you for your suggestions, which aim at strengthening our article. We found every single suggestion useful and therefore, we made an effort to implement all suggested changes (please find table below). In the journal’s guidance, there was a note, asking us to highlight our changes and revisions in the analysis. However, with the additions of news parts; extensive editing throughout the script; and proofreading at the end, almost every single line had to be highlighted. Therefore, instead of highlighting changes, we decided to submit a table with thorough and detailed explanations of what we did.
Once again, thank you very much for accepting to evaluate our revised work. If any other changes are needed, we are more than happy to work on the paper.
We are much obliged for your kind input.
Cordially,
Key tables must be improved to be visible and presented clearly. Until this, I can't recommend it for publishing.
|
Thank you, this is a very important comment. All Tables have been updated, using the same font style, size and are now clearer. |
Reviewer 3 Report
Comments and Suggestions for Authors
Overall, I feel the article has good potential. However, in its present form, it cannot be acceptable for publication due to several issues, which I shall elucidate.
* The abstract can be condensed and written more clearly. Please specifically state the two studies conducted, in addition to providing an overview of the results.
* Figure 1 - do the blue and grey arrows depict the same interaction? In that case, remove the blue arrows. If they are not the same, please explain what they depict.
* Between lines 35 and 45, it is essential to discuss the types of cultural heritage i.e. tangible and intangible.
* Between lines 46 and 59, several statements are made without citations. These are big claims, so citations are a must.
* What is ESD in line 53?
* Lines 125 to 127 - This is a big claim. Please refer to the following studies on cultural heritage informatics:
Modrow, S., & Youngman, T. (2023). Theorizing Cultural Heritage Informatics as the Intersection of Heritage, Memory, and Information. Proceedings of the Association for Information Science and Technology, 60(1), 666-671.
Koya, K., & Chowdhury, G. (2020). Cultural heritage information practices and iSchools education for achieving sustainable development. Journal of the Association for Information Science and Technology, 71(6), 696-710.
* There should be more literature stating the relationship between digital skills, cultural heritage and sustainable development/education.
* Lines 145 to 148 - This claim needs more backing from the literature or should be avoided altogether until the results are presented and discussed.
* It is important you state the overall research aim and objectives towards the end of the literature review to clarify what research questions you are trying to address.
Methodology:
* Was ethical approval sought from the authors' institutions? If so, please include a statement.
* Why were Cyprus, Lithuania and Italy chosen?
* Please attach the questionnaire as part of the appendices.
* How were the questionnaire responses analysed?
* Was informed consent sought from the participants?
* Why weren't the questionnaire respondents invited to participate in the interviews, as I would imagine they form a stakeholder category too?
* Were the interviews transcribed? Was a data analysis tool i.e. Nvivo utilised?
* Please explain in detail how the conversation/content analysis was applied - how was coding performed, when did code saturation occur, how were the themes defined, theme interpretation and consistency.
Results:
* Descriptive stats are well discussed.
* Section 3.2 and table 4 - How are you able to say there is a lack of awareness of tangible and intangible cultural heritage when options 5, 4 and 3 are the most selected for this part.
* There needs to be some form of statistical testing to determine a statistically significant relationship, to come to these conclusions.
* It is not entirely clear from the findings section if all the results from the questionnaire and the interviews are discussed. Therefore, the conclusions cannot be vouched.
* Within the discussion section, it is good to see Kolb's learning cycle utilised for reflexivity in developing a novel framework, however, the data analysis and results section needs to be solidified first. Without substantial and verifiable results, a novel framework cannot be developed.
Comments on the Quality of English LanguageThe article needs extensive editing and structuring to improve it's comprehensibility. Additionally, there are several instances of spelling errors that need to be corrected.
Author Response
Dear Reviewer,
Thank you for your suggestions, which aim at strengthening our article. We found every single suggestion useful and therefore, we made an effort to implement all suggested changes (please find table below). In the journal’s guidance, there was a note, asking us to highlight our changes and revisions in the analysis. However, with the additions of news parts; extensive editing throughout the script; and proofreading at the end, almost every single line had to be highlighted. Therefore, instead of highlighting changes, we decided to submit a table with thorough and detailed explanations of what we did.
Once again, thank you very much for accepting to evaluate our revised work. If any other changes are needed, we are more than happy to work on the paper.
We are much obliged for your kind input.
Cordially,
Reviewer 3 |
Actions |
The abstract can be condensed and written more clearly. Please specifically state the two studies conducted, in addition to providing an overview of the results. |
Thank you kindly for this suggestion. The Abstract has been restructured and rewritten to remove repetitive material. We also made the methods and findings clearer. |
Figure 1 - do the blue and grey arrows depict the same interaction? In that case, remove the blue arrows. If they are not the same, please explain what they depict. |
Many thanks for this comment, which enables us to unpack the content of the figure and explain it further. We added a relevant explanation in the analysis as follows: Moreover, as presented in Figure 1, the links between Sustainable Education, Skills and Cultural Heritage are further enhanced by the increasing adoption of digital technology in education [9, 53, 54, and 55]. The inner set of arrows in the figure represent the interconnection and interdependence among the different components towards sustainability. The outer set of arrows exists in the periphery of interdependence, and reveals a dynamic in which the three components are further enhanced and reinforced, as part of perpetual cycle of sustainability. |
* Between lines 35 and 45, it is essential to discuss the types of cultural heritage i.e. tangible and intangible. |
Thank you for this suggestion. A relevant explanation has been added in lines 45-51. |
* Between lines 46 and 59, several statements are made without citations. These are big claims, so citations are a must. |
Citations have been provided. Thank you. |
* What is ESD in line 53? |
The abbreviation has been corrected. Than you. |
* Lines 125 to 127 - This is a big claim. Please refer to the following studies on cultural heritage informatics: Modrow, S., & Youngman, T. (2023). Theorizing Cultural Heritage Informatics as the Intersection of Heritage, Memory, and Information. Proceedings of the Association for Information Science and Technology, 60(1), 666-671.
Koya, K., & Chowdhury, G. (2020). Cultural heritage information practices and iSchools education for achieving sustainable development. Journal of the Association for Information Science and Technology, 71(6), 696-710. |
Many thanks for providing such valuable references, which aim at strengthening readers’ understanding. Both references have been added in the analysis. |
There should be more literature stating the relationship between digital skills, cultural heritage and sustainable development/education. |
Thank you. Three (3) new references have been added (53,54,55) |
Lines 145 to 148 - This claim needs more backing from the literature or should be avoided altogether until the results are presented and discussed |
Many thanks for this comment. The specific sentence has been removed. |
Methodology:
* Was ethical approval sought from the authors' institutions? If so, please include a statement.
* Why were Cyprus, Lithuania and Italy chosen?
* Please attach the questionnaire as part of the appendices.
* How were the questionnaire responses analysed?
* Was informed consent sought from the participants?
* Why weren't the questionnaire respondents invited to participate in the interviews, as I would imagine they form a stakeholder category too?
* Were the interviews transcribed? Was a data analysis tool i.e. Nvivo utilised?
* Please explain in detail how the conversation/content analysis was applied - how was coding performed, when did code saturation occur, how were the themes defined, theme interpretation and consistency. |
Thank you for these comments.
No qualitative content analysis was applied for the literature review. Therefore, this phrase has been removed. What we meant is that we applied coding and categorization of qualitative findings. To do so, we applied a two-cycle coding. A relevant explanation has been added in the analysis.
Data saturation was never reached; and saturation has never been the purpose. It was very unlikely to reach homogeneity and repetitiveness in findings since the stakeholders participating in the interviews represent different interests and needs. Rather, the aim was to grasp the opinion of different authorities, who have a role to play at the intersection of cultural heritage, digitization and education.
Since the study was undertaken as part of an EU-funded project, a research ethics’ approval was granted by the project-consortium’s legal office. A relevant explanation has been added in the analysis.
The collection of findings through interviews and questionnaires applied international research ethics principles and tools, including informed consent, anonymity, confidentiality and data privacy. Participants (for interviews) and responders (for questionnaires) were provided with a cover letter, explaining the aims and objectives of the study, along with potential benefits for education. A relevant exmpanation has been added in the analysis.
Questionnaires’ responses were analysed through SPSS. Relevant explanation has been in the analysis. Also, the questionnaire has been added as an Appendix.
The 3 countries participating in the study (Cyprus, Lithuania and Italy) have been selected due to their similar percentage of dropout rates. That was a striking similarity among the three countries whereas, the study’s aim was to examine the countries with low digital literacy and high percentage of drop out. A relevant section has been added in the analysis.
The interviews’ participants were key stakeholder-representatives, including museums officers, policy-makers, and education authorities. The fieldwork’s aim was to grasp the opinion of different authorities who have a role to play at the intersection of cultural heritage, digitization and education. Relevant explanation has been added in the analysis.
Interviews have been analysed manually through a two-cycle coding. The first cycle of coding was undertaken immediately after each interview, while the discussion’s specifics were still fresh. The first cycle included ‘categorising and labelling managerial responses and matching recurring opinions under key themes’ [62]. This process was not about adding names and categories to the data record, but also to interpret expressions and synthesise small sentences, which then became paragraphs. During this phase, new categories emerged, since the findings were often new or unexpected. Then, the second-cycle coding enabled researches to re-analyse, re-organise, and re-synthesise the material produced through first-cycle coding, towards producing bigger chunks of analysis. Relevant explanation has been added in the analysis.
Kindly note: following yours and other reviewers’ feedback, we added many other details explaining the research design, including sampling approach, sampling frame, saturation et cetera. The ‘Methodology’ section has been restructured entirely. Thank you
|
Results:
* Descriptive stats are well discussed.
* Section 3.2 and table 4 - How are you able to say there is a lack of awareness of tangible and intangible cultural heritage when options 5, 4 and 3 are the most selected for this part.
* It is not entirely clear from the findings section if all the results from the questionnaire and the interviews are discussed. Therefore, the conclusions cannot be vouched.
* Within the discussion section, it is good to see Kolb's learning cycle utilised for reflexivity in developing a novel framework, however, the data analysis and results section needs to be solidified first. Without substantial and verifiable results, a novel framework cannot be developed. |
Thank you for this comment. A more extensive explanation has been added.
We confirm that all findings are discussed in the analysis.
Thank you very much for this valuable suggestion. Further analysis has been attempted, linking Kolb’s learning cycle with reflexivity. |
The article needs extensive editing and structuring to improve it's comprehensibility. Additionally, there are several instances of spelling errors that need to be corrected. |
The article has been edited throughout. Then it has been sent for professional proofreading. |
Round 2
Reviewer 1 Report
Comments and Suggestions for Authors
I am pleased with the authors modifications to the original manuscript. My concerns were addressed and thoroughly explored. This piece is worthy of publication.
Comments on the Quality of English LanguageEnglish language is easy to follow and understand. I only found minor errors, most of them were punctuation errors.
Author Response
Dear Reviewer,
Thank you ever so much for your guidance throughout the process.
Cordially,
Authors
Reviewer 2 Report
Comments and Suggestions for Authors
Dear Authors,
I can't recommend publishing, since there are a statement without any proof and obviously wrong statistical conclusion.
All my comments are within the revised manuscript version attached below.
There are only a few comments in the file attached.
Author Response
Dear Reviewer,
Thank you ever so much for your valuable feedback. We really appreciate the time you took to write direct comments for revisions on our paper. Although we earlier conducted professional proofreading (and paid for it), your review enabled us to correct additional typos and discrepancies. We also changed digitization to digitilisation (as you suggested) on the graph and throughout the analysis. We also removed repetitive sentences, as you suggested.
The only suggestion we didn’t change concerns a sentence in the introduction, which aims at introducing the reader to cultural heritage. Although you suggested that the particular sentence may be unnecessary, we decided to keep it to enable 1st time readers to formulate an understanding about the components of cultural heritage (tangible and intangible).
Moreover, concerning your comment on statistics (presented in section 3.1), it seems that the table (Table 1) supporting the analysis was wrong. In the actual script on the platform, the correct ‘Table 1’ is visible in the analysis. However, on the version where you added direct comments, Table 1 is the same with Table 2. We don’t know how this problem occurred, possibly during the transfer of infographics. But in any case, please let us apologise for this misunderstanding. We now have the correct Table 1 on the revised script, which presents all correct figures. We also added the actual figures in the analysis, to enhance the analysis further.
We also edited the analysis further and added another figure, explaining our coding process.
Also, kindly find the questionnaire attached. The journal's team will place it within the actual paper. But you can also find it here as an attachment.
We remain at your disposal for any other changes you consider necessary.
Thank you very much for your support.
Cordially,
Authors

Reviewer 3 Report
Comments and Suggestions for Authors
* The example provided between lines 126-132 is not relevant to the gist of the article. Please include a suitable example.
*The coding part of the data analysis is still not clear. Which criteria was used? Why did coding begin separately for every interview? For example https://images.app.goo.gl/8UwJUYQwEEmeZPrw6 for reference.
*Could you also confirm if the interviews were conducted in home languages (or English) and how was consistency in findings was achieved?
Comments on the Quality of English Language
The article has been improved significantly, however, I suggest a further round of proofreading to improve readability.
Author Response
Dear Reviewer,
Thank you ever so much for your valuable feedback. We really appreciate the time you took to suggest specific changes and revisions.
Change 1
Comment: ‘The example provided between lines 126-132 is not relevant to the gist of the article. Please include a suitable example’.
Answer: you are absolutely right. This example is completely irrelevant. The you very much for suggesting it. It has been replaced with a revenant example and references as follows:
For example Borthwick et al. [51] and Kumar et al. [52] state that ‘[l]earners can down-load the necessary information or upload their content using a plethora of digital resources’. Web 2.0 tools (wikis, podcasts, blogs, and so on) enable learners to create material, collaborate with others, evaluate each other's work, and progress toward co-learning.
Change 2
Comment: ‘The coding part of the data analysis is still not clear. Which criteria was used? Why did coding begin separately for every interview?’.
Answer: many thanks for this comment, which aims at strengthening the analysis. The main codes were comprised by the competencies appearing in Table. Further explanation has been added, along with a new figure (similar to the example you have provided us with – many thanks for this).
Change 3
Many thanks for the comment on interviews (Could you also confirm if the interviews were conducted in home languages (or English) and how was consistency in findings was achieved?).
Answer: A relevant explanation has been added in the analysis, as follows:
The interviews were conducted in home languages, and translated into English (which is the projects official language) by the project designated translators.
We remain at your disposal for any other changes you consider necessary.
Thank you very much for your support.
Cordially,
Authors
Round 3
Reviewer 2 Report
Comments and Suggestions for Authors
Dear Authors,
If you declare that the Appendix is added, then the Appendix must be added.
For now, I can't recommend the manuscript to be published.

Author Response
Dear Reviewer,
Many thanks for your valuable feedback and direct comments in the script. We conducted all the changes you suggested in the script – all of them were meaningful corrections, thank you.
Concerning the appendix, kindly note that we received a message from the Editorial office that the Appendix was formatted as Supplementary Material (not as Appendix), and it’s available. Therefore, we removed an points referring to an appendix in the script.
We remain at your disposal for any other changes you consider necessary.
Thank you very much for your guidance.
Cordially,
Authors
Reviewer 3 Report
Comments and Suggestions for Authors
Thank you for the amendments following the feedback.
Comments on the Quality of English LanguageNA
Author Response
Dear Reviewer,
Many thanks for your positive response. We will now conduct a final copy-edit and submit the final version.
We really appreciate your contribution to the evaluation process.
Cordially,
Authors.